# Investigating the Prevalence and Predictors of Injury Occurrence in Competitive Hip Hop Dancers: Prospective Analysis

**DOI:** 10.3390/ijerph16173214

**Published:** 2019-09-03

**Authors:** Eva Ursej, Damir Sekulic, Dasa Prus, Goran Gabrilo, Petra Zaletel

**Affiliations:** 1Faculty of Sport, University of Ljubljana 1000 Ljubljana, Slovenia; 2Faculty of Kinesiology, University of Split, 21000 Split, Croatia

**Keywords:** sport, prevalence, risk factors, protective factors, OSTRC

## Abstract

Hip hop is a popular form of competitive and recreational sport worldwide, but studies rarely investigate injury prevalence and factors associated with injury occurrence in this sport. This study aimed to prospectively examine injury occurrence in hip hop dancers in a three-month period and to evaluate potential predictors of injury occurrence in hip hop dancers. The participants were 129 competitive hip hop dancers (114 females, 17.95 ± 4.15 years of age). Study predictors were obtained at study baseline and included sociodemographic factors, sport-related factors, previous injury status, anthropometric and body build indices (body height, mass, body mass index, and body composition variables), and dynamic balance performance (obtained by the Star Excursion Balance Test—SEBT). The outcome was injury occurrence, which was prospectively observed once a week by the Oslo Sports Trauma Research Center Overuse Injury Questionnaire (OSTRC). During the course of the study, 101 injuries occurred, equating to an annual injury incidence of 312%. On average, each dancer suffered 0.78 injuries (95% Confidence Interval (95% CI): 0.61–0.97) across a study period of three months (0.76 (95% CI: 0.60–0.95) and 0.93 (95% CI: 0.75–1.13), in females and males, respectively; Mann Whitney Z-value: 0.68, *p* = 0.52). Seventeen percent of dancers suffered multiple injuries, and 49% of all injuries were time-loss injuries. The knee was the most frequently injured body location (42% of all reported injuries), followed by the back region (32%) and the ankle (15%). Previous injury was a strong predictor of injury occurrence (Odds Ratio: 3.76, 95% CI: 1.87–4.59). Lower injury risk was evidenced among those participants who achieved better scores on several SEBT variables, irrespective of gender and previous injury status; with no significant influence of anthropometric and body build variables on injury occurrence. This study highlighted a high injury rate in hip hop dancers. Dancers and coaches should be informed about the certain protective effects of dynamic balance on the prevention of musculoskeletal injury in hip hop in order to assure safe and effective practices. The usage of SEBT as a convenient and cheap testing procedure is encouraged in other dance disciplines.

## 1. Introduction

Hip hop dance refers to street dance styles performed mostly to hip hop music; it includes a wide range of styles, primarily “breaking”, and was created in the late 1960s and early 1970s in the Bronx, New York. From its very beginning, hip hop dance has evolved rapidly and now includes many different dance styles, generally divided into “old school” (i.e., breaking, popping, locking), and “new school” (i.e., krumping, Harlem shake, house, street jazz). Each of them puts dancers into different situations, demanding a wide spectrum of physical abilities and tricks [1,2]. With the increasing popularity of this sport, there is a growing body of literature directly focused on the acute or chronic effects of hip hop dance [3,4,5,6,7,8], performance-related factors [9,10,11,12], and different health-related problems associated with hip hop dance culture, including injury occurrence [1,13,14,15,16]. With the increase of popularity (and more competitors), dance choreography is becoming more complex, which results in an increase of the physical demands of hip hop dance, higher aerobic and anaerobic fitness, muscular strength, agility, coordination, and necessity for better motor control [17]. While dancers regularly perform rapid and repeated motions with muscles at their functional disadvantage, it altogether leads to specific injuries that are basically related to the specificity of musculoskeletal (MSK) load [13].

The identification of injury occurrence and factors associated with injury occurrence in any type of physical activity is an important issue in the development of the environment, which will allow safe and effective practice and the possible development of specific preventive programs [18,19,20,21]. However, despite the clear necessity for more detailed identification of MSK problems in hip hop, and apart from several clinical case studies where authors presented specific clinically treated injuries and other health-related problems (i.e., C1/C2 subluxation, avulsion of the anterior superior iliac spine, scaphoid nonunion), there is an evident lack of studies that systematically investigated the prevalence of MSK injuries in hip hop [14,15,22].

In one of the first epidemiological studies of this problem, Cho et al. retrospectively examined injury rates and patterns among 23 professional and 19 amateur breakdancers [23]. In short, the authors reported 95.2% MSK injuries, with the wrist (69.0%), finger (61.9%), and knee (61.9%) being the most injured locations. Furthermore, sprains, strains, and tendinitis were the most common type of injuries [23]. In another retrospective study, Kauther et al. surveyed 40 breakdance professionals and 104 amateurs [24]. The results showed significantly more injuries and overuse syndromes in the professionals, with approximately one injury every 228.6 h. Furthermore, the authors highlighted that even with severe injuries, dancers rarely and only very briefly interrupt their training [24]. Finally, Ojofeitimi et al. presented results of a web-based survey of 232 dancers (13+ years of age) and reported a total of 738 injuries. Approximately 70% were time-loss injuries, and specific figures were reported for different hip hop styles. The highest injury rate was found for breakdancers, followed by poppers/lockers and new schoolers (3.8, 2.3, and 2.3 injuries per injured dancer, respectively) [1].

Compared with other dancers (modern, ballet, tap, aerobic), hip hop dancers experience a high injury incidence. Specifically, the annual injury incidence for hip hop dancers was 3.4 times the rate for modern dancers and almost twice as high as the injury rate recently reported for professional ballet dancers [1,25]. For a simple comparison, this is an injury incidence similar to that reported for artistic gymnasts, one of the competitive sports with the highest injury risk [1]. Despite such high injury occurrence, only one study investigated some factors as possible correlates of injury in hip hop, and the authors found no significant association among performance level, the usage of protective devices, the inclusion of warm-up, and injury occurrence [23]. Meanwhile, no study so far has examined other potentially important factors, such as anthropometric and body build indices or motor capacities, which may potentially be related to injury occurrence in this sport. Specifically, hip hop dancers perform their movements while conquering their own body, with muscles at a functional disadvantage. Therefore, it is logical to expect that body dimensions (i.e., anthropometrics) and characteristics of body build may present as certain factors of influence on injury occurrence, as has been reported for other dance activities [26,27]. Additionally, the performance of hip hop dancers is directly influenced by their balance capacities, which are specifically challenged during different routines, but no study so far has examined balance as a potential predictor of injury status in this sport. This is additionally important because studies from other sports have confirmed the importance of balance capacity in predicting injuries of the lower extremities, which are known to be the most common injured locations, even in hip hop dance [1,28,29,30].

From the previous literature overview, it is clear that despite the evident injury risk in hip hop dance, there is a lack of information on factors that may be associated with injury occurrence in this sport. Additionally, all previous studies that investigated the injury occurrence in this sport were retrospective, and the authors themselves clearly highlighted the necessity of prospective analyses of this problem [1,23,24]. Therefore, the aim of this study was to prospectively examine injury occurrence in hip hop dancers during a three-month period and to evaluate dance-related variables, anthropometrics/body build indices, and dynamic balance performance as specific predictors of injury occurrence in hip hop dancers. Initially, we hypothesized that the studied predictors will be significantly correlated with injury occurrence in the studied dancers. The identification of factors influencing injury occurrence will allow for the development of accurate intervention strategies to mitigate the injury problem in this sport.

## 2. Methods

### 2.1. Participants

The participants in this study were hip hop dancers from Slovenia (*n* = 129, 114 females, 17.95 ± 4.15 years of age). The selection of participants was performed between June and August 2018 through direct contacts with dance teachers who received detailed information about the study purpose and procedures. They informed their students, and those willing to be enrolled were recruited through invitations to the dance school during September and October 2018. Hip hop dancers were considered eligible if they were (i) aged at least 14 years; (ii) competitive in hip hop dance and were officially registered at the national dance federation as competitors; and (iii) were healthy and uninjured. At study baseline, there were 22 dance schools with registered dancers older than 14 years of age in the country, with 345 dancers older than 14 years. Then, 50% of the schools were randomly selected and invited to participate in the study (e.g., 11 schools with 189 dancers older than 14 years). The participants were informed that they could refuse participation and withdraw from the study at any time and for any reason, and their informed written consent was obtained. Participation in the study was voluntary and anonymous, and no personal data from the participants were included in the study. The study protocol was approved by the Ethics Committee of the University of Ljubljana, Faculty of Sport, Ljubljana, Slovenia (Ref. number: 1175/2017).

### 2.2. Variables and Measurement

The variables in this study included sociodemographic factors (e.g., gender, age), sport-related factors (dance factors), anthropometric and body build features, balance performance (all observed as predictors), and outcome - injury status. The predictors were generally measured at baseline, while the number and duration of training sessions, as well as the outcome (injury status) were evaluated once a week during the study course of 15 weeks.

Throughout dance factors, participants were asked about their (i) experience in hip hop dance (in years), (ii) age when they started to practice hip hop (later transformed into years of experience in hip hop), (iii) number of training sessions per week, (iv) hours of training per week, (v) number of competitions per month, and (vi) current dance score (based on their success (results) in the last two competition seasons, ranging from 1 (Finals at the European and World Championships) to 4 (participation at the national championship)).

Anthropometric variables included body height (measured while the participant was standing erect against a portable stadiometer without shoes, in 0.1 cm increments), body mass (in 0.1 kg increments), calculated body mass index (BMI; in kg/m^2^), and body composition indices (body fat percentage (BF%), skeletal lean mass (kg), and fat free mass (kg)), which were measured by bioelectrical impedance analysis with the InBody 720 Tetrapolar 8-Point Tactile Electrode System (Biospace Co. Ltd., Seoul, Korea). In brief, this device measures and analyzes an individual’s body composition (i.e., water, body fat, protein, muscle, and bone minerals) and determines the weight of lean muscle tissue in each limb, the body’s water content, body fat percentage (BF%), bone mineral content, protein content, and visceral fat levels. The system measures resistance in broadband frequencies (1, 5, 50, 250, 500, and 1000 kHz) and reactance in mean frequencies (5, 50, and 250 kHz), and for this investigation, the testing was performed with an alternating current of 90 μA (1 kHz) and 400 μA. During the measurement, participants placed bare feet on the metal plates of the system, confidently grasped the grips, and placed all fingers at the standardized locations. During the procedure, the participants held out their arms and legs so that they would not come into contact with any other body segments. The measurement required no specific skill and took approximately 2 min per participant with “research mode” activated. This InBody 720 was extensively studied with regard to its reliability and validity, and reports showed appropriate to high metric characteristics in various samples, including physically active individuals and athletes [31,32,33].

Balance performance was measured by the Star Excursion Balance Test (SEBT), which is often used to assess dynamic balance and screen deficits in dynamic postural control due to MSK injuries (Figure 1) [34,35]. Studies conducted so far established a comprehensive portfolio of validity for the SEBT, and it is considered a representative and reliable non-instrumented dynamic balance test for physically active individuals. Additionally, the SEBT has been previously shown to be a reliable measure and has validity as a dynamic test to predict the risk of lower extremity injury [36]. It is composed of eight line grids, extending from the center at 45 degrees, that are taped on the floor with adhesive tape marked with centimeters. The SEBT consisted of a series of single-limb squats, where participants stood barefoot with the stance foot at the center of the grid (Figure 1A). While maintaining a single-leg stance, dancers had to reach along the marked line as far as possible, using the opposite leg, and then return the reach leg back to the center of the grid, without losing their balance. During the test, dancers held their hands at the iliac crest. The test was not counted if (i) the dancer was not able to maintain the single-leg stance, (ii) the dancer lifted their standing leg off the floor during the trial, (iii) the dancers’ weight was transferred onto the reaching foot, (iv) the dancer’s hands did not remain on their hips, or (v) the dancer was not able to firmly maintain the start and return position. The test was verbally explained and demonstrated by the examiner, who then oversaw the proper execution of the test. The reach distances were read with centimeter accuracy and normalized to the % leg length of the participants. The variables observed in this investigation included normalized SEBT performances when participants were standing on the right leg (SEBT_R1–SEBT_R8), and on the left leg (SEBT_L1–SEBT_L8) (Figure 1B).

In this study, the injuries were recorded with the Oslo Sports Trauma Research Center Overuse Injury Questionnaire (OSTRC) [37,38,39]. Participants responded to the OSTRC at study baseline and prospectively once a week over the course of the study. At baseline, participants were asked about injury occurrence in the period of 6 months before the testing. The digital form of the questionnaire was sent to participants by e-mail once a week. Additional individual reminders were sent to the participants who did not provide any data for the preceding week. Personal phone calls were made if the participants did not react to the email reminders and if the reported information on the injury form was found to be inconsistent with previous records. The outcome in this study was the incidence of ankle, knee, back, and shoulder problems and injuries that occurred from baseline to 15 weeks of follow-up. Each answer in the OSTRC corresponded to a score, and for each question, a score between 0 and 25 was given, which lead to a sum score between 0 and 100 for the total questionnaire. Scores of 40 or more reported during the follow-up were classified as an MSK-injury occurrence (MSK-injury; later numerically scored as “2” for the purpose of the multinomial regression analyses, please see Statistics subsection for details). If the participant scored anything higher than the lowest grades on each question, the score was categorized as the presence of an MSK problem (MSK-problem; scored as “1” in regression calculation). Finally, if the participant reported the lowest grades for all questions, the absence of any problem/injury was recorded (scored as “0” in regression).

### 2.3. Statistics

The normality of the distributions was checked by the Kolmogorov–Smirnov test. For normally distributed variables, descriptive statistics included means and standard deviations. Injury rates were reported as the total number of injuries per studied period, and the number of injuries relative to hours of exposure—dance hours (with 95% CI for Poisson rates). Irrespective of OSTRC-specific graduation, which differentiates MS-problem from MS-injury, in the following text, the MS-problem (OSTRC scores from 1 to 40) and MS-injury (scores of >40) were collectively considered as “injury” if not specified otherwise.

The differences between genders in studied factors and injury occurrence were evaluated by the chi-square (for nominal variables) and Mann–Whitney (for ordinal variables) tests.

The association between the studied predictors and outcome (OSTRC-defined presence of MSK-problem and MSK-injury) was evaluated by univariate multinomial regression calculation for a multinomial criterion based on the OSTRC scale (0—absence of MSK problem/injury, 1—MSK-problem, 2—MSK-injury), with the absence of a problem/injury as the referent value in the multinomial regression calculation. The odds ratio (OR) with the corresponding 95% confidence interval (95% CI) was reported. Multinomial regression calculations included non-adjusted regression correlations, and correlations adjusted for gender and injury reported at baseline.

Statistica ver. 13.5 (Tibco Inc., Palo Alto, CA, USA) was used for all analyses, and a significance level of *p* < 0.05 was applied.

## 3. Results

The descriptive statistics for studied dance-related factors, anthropometric and body build variables, and SEBT scores are presented in Table 1.

During the course of the study, 64 dancers reported at least one injury, which resulted in prevalence of 50%. Altogether, 101 injuries occurred (95% CI: 82–122), over approximately 10,400 h of exposure (90 h of exposure per dancer). On average, each dancer suffered 0.78 injuries (95% CI: 0.61–0.97) across a study period of 3 months (0.76 (95% CI: 0.60–0.95) and 0.93 (95% CI: 0.75–1.13) in females and males, respectively), with no significant difference between genders (Mann–Whitney Z-value: 0.68, *p* = 0.52). It equates to a rate of 3.12 injuries per dancer per year (e.g., annual incidence of 12%).

When expressed with regard to exposure time, i.e., hours of dance (HD), the prevalence was 8.7 injuries per 1000 HD (95% CI: 6.9–1.1), with somewhat higher prevalence in males than in females (males: 10.33 (95% CI: 5.1–14.2), females: 8.44 (95% CI: 3.5–15.1)). However, it must be noted that this prevalence included MSK-problems and injuries (see previous explanation about OSTRC as measuring scale).

The 17% of dancers reported multiple injuries. Specifically, 10% of dancers reported two, 6% reported three, and 1 dancer (less than 1%) reported four injuries. The 49% of all injuries were time-loss injuries (47% and 55% of time loss injuries in females and males, respectively).

The knee was the most frequently injured body location (43% of all reported injuries), then the back region (33% of all injuries), and ankle (15%), with similar prevalence across genders (Chi square: 1.17, *p* = 0.75) (Table 2).

When separating MSK-injuries (OSTRC scores >40) from MSK-problems (OSTRC: 1–40), the 17% of dancers suffered an MSK-injury, while an additional 33% reported MSK-problems, with no significant difference between genders (Chi square: 0.62, *p* = 0.73).

Injury that occurred during the period of 6-months before the study baseline was a strong predictor of MSK-injury and MSK-problem over the course of the study. Specifically, previous injury increased the risk for occurrence of MSK-problem over the course of the study by more than 2.5 times (OR: 2.62, 95% CI: 1.61–4.58), and risk for occurrence of MSK-injury by more than 3.5 times (OR: 3.76, 95% CI: 1.87–4.59).

The SEBT measures obtained at study baseline were significantly associated with MSK-problem and MSK-injury over the study course. Specifically, crude multinomial regressions evidenced lower risk for occurrence of MSK-injury in participants who achieved better results in several SEBT performances, specifically: SEBT_R4 (OR: 0.95, 95% CI: 0.92–0.99), SEBT_R5 (OR: 0.94, 95% CI: 0.92–0.97), SEBT_L3 (OR: 0.95, 95% CI: 0.91–0.99), SEBT_L4 (OR: 0.95, 95% CI: 0.92–0.99), SEBT_L5 (OR: 0.96, 95% CI: 0.93–0.99), and SEBT_L6 (OR: 0.96, 95% CI: 0.93–0.99). Furthermore, participants with better SEBT_R6 and SEBT_L7 had lower risk for reporting MSK-problem (OR: 0.95, 95% CI: 0.91–0.97, and OR: 0.98, 95% CI: 0.96–0.99, respectively) (Table 3).

When multinomial regressions for injury occurrence were calculated while controlling gender and previous injury as covariates (Model 3), lower risk for MSK-injury during the course of the study was found in those participants who had better scores at SEBT_L3 (OR: 0.96, 95% CI: 0.92–0.99), SEBT_L5 (OR: 0.96, 95% CI: 0.93–0.98), and SEBT_L6 (OR: 0.97, 95% CI: 0.93–0.99) (Table 4).

## 4. Discussion

This study aimed to prospectively evaluate injury occurrence and specific factors associated with injury occurrence in hip hop dancers. In general, the prevalence of injury is high, and we evidenced some specific correlates of injury occurrence. The most important findings will be discussed in the following text.

### 4.1. Injury Occurrence in Hip Hop Dancers

Altogether, our results of injury occurrence correspond to the occurrence of approximately 3.1 MSK injuries/problems per dancer per year (annual injury incidence of 310%). Therefore, our results showed higher injury risk than previous studies where authors reported annual injury risk of 237% in hip hop dancers [1]. However, this difference is probably a result of: (i) Different study designs (i.e., our study was prospective, while the previous one was retrospective) and consequent possibility of recall bias, and (ii) the fact that, in this study, we used OSTRC as a specific measurement tool [1,40].

The knee was the most injured location both in females and in males, accounting for approximately 43% of all injuries, followed by lower back injuries (33% of all injuries). Although herein we did not specifically examine the mechanisms of the injury occurrence, the authors of the study are deeply involved in hip hop dance as coaches and former dancers and therefore may discuss the most likely background and mechanisms of the injuries from such perspectives. Briefly, hip hop dancing techniques include fast and complex footwork, deep squats, twists, and quick changes in the direction of movement, most of the time in unnatural body positions. An important part of the choreography is acrobatic elements, where dancers hit or collide with the ground with great force [24]. All those rapid movements, rotations, and deviations in flexed and extended knee joints with jumping and landing present a high risk of knee injury in hip hop dancers.

With regard to lower back as the second most injured location (accounting for 33% of all injuries), the following mechanisms of occurrence should be considered. The combination of footwork and fluidity of movement in the upper body involves core muscle and trunk motor control, while turns and jumps require proper core stabilization [2]. It is clear that positions obtained during such performances are hardly to be observed as anatomically natural or functional. Consequently, such unnatural body positions with quick changes in movements and intensive rotations increase the forces that affect the lower back. The literature suggests that the most common cause of lower back injury is trunk muscle imbalance [41], which can lead to improper muscle activation during repetitive, high-impact activities, such as those that appear throughout the hip hop dancing performance.

As the most mobile joint in the human body, the shoulder joint is very prone to injuries because of its great instability. The anatomy and physiology of shoulder movement tell us that the motion is largely dependent on soft tissues, which are impacted by great forces during hand-weight-bearing choreography and rotations in closed kinetic chain moves in hip hop dance. The higher prevalence of shoulder injuries in males than in females (20% and 9% of all injuries for males and females, respectively) is actually related to techniques in dancing that are known to be more common in males than in females. Specifically, all power moves that are part of dynamic handsprings and spinning are mostly performed by males and put the upper extremities, specifically the shoulders, at risk for injury [23].

### 4.2. Correlates of Injury Occurrence in Hip Hop Dancers

One of the most important findings of this study is related to the relationship between balance capacity and injury risk in hip hop dancers. In short, dancers with better balance were less likely to suffer injury during the observed period of time. Although the clinical importance of these findings is probably relatively small, this is one of the first studies that prospectively observed balance as a predictor of injury in dance, and from our perspective, these results deserve attention. Therefore, a short overview of the physiological background is needed. Postural balance could be defined as the ability to achieve a status of equilibrium by maintaining the body’s center of gravity over the base of support [30]. The control of balance involves a continuous feedback system of processing visual, somatosensory, and vestibular inputs and executing (appropriate) neuromuscular actions. An important component of the somatosensory system that is directly responsible for balance is proprioception (the ability to sense stimuli arising within the body regarding position, motion, and equilibrium, and afferent information on position and movement coming from internal body receptors in joints, tendons, and muscles) [30,42].

The importance of balance in injury risk is a known issue in sports and exercise, and studies have frequently confirmed poor balance ability as a risk factor for injury occurrence in a number of sports [30]. However, for the purpose of this study, the most interesting findings are from investigations where authors used the SEBT and its variations [43]. In short, the SEBT and its modifications were observed as being related to injury occurrence in US college athletes [43,44] and high school basketball players [28,29]. Meanwhile, although some authors recognized the potential applicability of the SEBT in dance [34,45], to the best of our knowledge, no study so far has examined the validity of this test for the prediction of injury in any form of dance.

Previous brief presentation of the most common movements and routines in hip hop dance (please see first subheading of the Discussion where we explained the etiology of injury occurrence for different body regions) may accentuate the association between SEBT performances (i.e., balance capacity) and risk for injury. Namely, although all dancers possess a specific “dominant side” (i.e., lateral preference) and perform movements accordingly, the performances that include several dancers (e.g., duo, group, formation, production) frequently require moves to be executed even on the “nondominant” side. This clearly increases the risk for injury occurrence, especially in dancers with an increased “lateral bias” and consequently accentuated functional asymmetry [46]. This is particularly possible with regard to knee injury (the most common injured body location), since nearly all intensive rotations and jumps require adequate muscular strength. Additionally, during prolonged performances, the consequent fatigue reduces the functionality of the muscles involved in knee joint movement, making proprioception and balance crucial protective factors that may control excessive forces and eventually prevent consequent injury. In hip hop dance, this is additionally accentuated by the fact that movements are often inconsistent with optimal biomechanical function. While the cause of acute, traumatic injuries is usually well known and is most often a result of high-intensity forces of short duration, chronic injuries are usually caused by low-intensity forces of long duration [47]. A review of dance injuries revealed that dancers suffer more from overuse and chronic injuries [48], which can be connected to poor technique, strength, and balance. To the best of our knowledge, our study is the first to directly confirm this hypothesis in hip hop dancers.

With regard to correlations obtained between balance capacity and injury risk, another issue deserves attention. In particular, the methodological approach used in the study where previous injury status was used as a covariate in multinomial regression analysis actually confirmed that the SEBT is a valid predictor of injury occurrence in hip hop dance, irrespective of previous injury. This is particularly important because the SEBT is known to be both (i) a measure of dynamic balance and (ii) an indicator of deficits in dynamic postural control due to (previous or current) MSK injury [36]. Therefore, one can argue that the correlation between initial balance status and follow-up injury occurrence may be a result of the suppressor effect of previous injury (i.e., a previous injury negatively affects balance and increases the risk for injury occurrence in the future). Indeed, this is partially true (i.e., some significant predictors from the regression model unadjusted for “occurrence of previous injury” were insignificant in the adjusted model), but generally, we may highlight that SEBT performances should be observed as valid predictors of injury occurrence in hip hop dancers, irrespective of previous injury status.

It has already been reported that anthropometric features and body build indices correlate with injury occurrence in dance activities [26,27]. Although we initially hypothesized that body build and anthropometric features would significantly influence the injury occurrence in hip hop dancers, our results do not support this hypothesis. In brief, none of the anthropometric and body build features measured at study baseline were related to injury occurrence over the observed period of time. Although there is an evident lack of investigations that examined these issues in hip hop dance, we may briefly discuss some possible explanations of our findings. First, hip hop dance consists of various disciplines (please see the Introduction for details). While some disciplines, such as dance battles, are characterized by powerful and quick movements and rapid changes in positions, and should be generally considered as high-impact activities (especially because dancers must adapt moves and structures to the certain instrumental phrases). Other categories (i.e., duos, small groups) are more fluent, balanced, choreographed, and, therefore, less physically stressful [49]. In addition, the rules and characteristics of hip hop dance allow each dancer to develop his/her own dance technique among his/her capacities. Such diversity of styles and disciplines allows each dancer to choose the most proper style, which allows the dancer a safe and effective dance performance. Therefore, it is possible that the dancers studied here are somewhat in between the range of physiologic and morphologic values that do not present a risk factor for an injury.

### 4.3. Limitations and Strengths

The first study limitation is related to the unequal number of female and male dancers. Indeed, the relatively small number of males in the study limits the generalizability of the results. However, this is the situation in Slovenia where the sample was drawn from, and we did not aim to study the differences between genders. In addition, despite the small number of studied male dancers, we believe that the study highlighted some important characteristics of injury occurrence in males (i.e., shoulder as a specific location of injury). Therefore, in future studies, special attention should be placed on male hip hop dancers. Next, we used the OSTRC measurement tool, which examines injury occurrence at a limited number of body sites. Meanwhile, hip hop dancers may suffer injury at different sites to those evaluated through the OSTRC questionnaire. Finally, the study focused on a relatively short time period (e.g., 3 months); therefore, its objectivity may be questioned.

This is the first prospective study on injury problems and factors related to injury occurrence in hip hop dance. Therefore, the obtained results allow a clear identification of (i) injury occurrence (i.e., there is no recall bias) and (ii) causality between the investigated factors and injury occurrence in hip hop dancers. Additionally, to the best of our knowledge, this is the first study to investigate anthropometric and body build indices as potential predictors of injury in this sport and one of the first studies to investigate balance as a predictor of injury in any form of dance. Therefore, the findings will hopefully initiate further investigation of the problem.

## 5. Conclusions

Hip hop dance continues to enjoy global popularity, both as a competitive sport and as a valuable recreational activity. This study confirmed previous reports of a high injury rate in hip hop dancers. Therefore, findings about predictors of injury occurrence are particularly important in order to provide a safe environment for hip hop dance practice. In general, our initial study hypothesis on the significant influence of the studied predictors on injury occurrence was partially confirmed.

Apart from the logical and expected correlation between previous injury status and injury occurrence (e.g., a higher risk for injury in those dancers who suffered an injury previously), this study demonstrated a reduced risk for injury in dancers with better dynamic balance, irrespective of previous injury status. While the SEBT requires no special equipment and is generally cheap and not time consuming, we may highlight its applicability in the evaluation of balance in various dance activities and the potential applicability in identification of dancers at specific risk for injury occurrence. Furthermore, we studied high-level dancers who are regularly facing highly demanding practice and intensive routines. Consequently, study participants should be considered to be at high risk for injury occurrence. As a result, the SEBT values presented here can be used as certain normative values for future comparison.

Anthropometric and body build indices were not correlated with injury occurrence in the studied hip hop dancers. Most likely, the performance level of the participants did not allow the identification of the prognostic value of anthropometric and body build indices in the prediction of injury occurrence. Specifically, the dancers studied here were involved in hip hop for several years, and throughout this period, they were probably oriented toward a certain dancing style that is most suitable with regard to their body dimensions and body build. The hip hop characteristic of allowing different participants to choose an appropriate form of dance style is almost certainly one of the reasons for the growing popularity of this sport worldwide.

This study evaluated some specific predictors of injury in hip hop. In future studies, other potentially important factors of influence on injury occurrence should be evaluated. In doing so, special attention should be placed on those abilities and characteristics that could be potentially related to injury occurrence of some specific body regions (i.e., injured shoulder in male dancers, lower back in females). Furthermore, knowing the differences in dance styles, and physiological demands of various hip hop dance disciplines, the analyses of specific dance disciplines are warranted.

## Figures and Tables

**Figure 1 ijerph-16-03214-f001:**
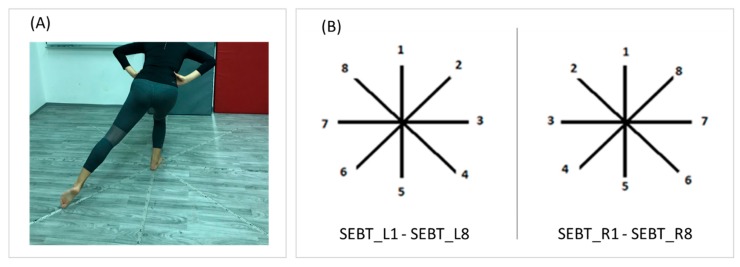
Star Excursion Balance Test (SEBT) execution (**A**), and SEBT scoring (**B**) while standing on the left leg (SEBT_L1–SEBT_L8), and while standing on the right leg (SEBT_R1–SEBT_R8) (B).

**Table 1 ijerph-16-03214-t001:** Descriptive statistics for studied dance factors, anthropometric and body build indices, and balance performance in hip hop dancers.

Variables	Total (*n* = 128)		Females (*n* = 113)		Males (*n* = 15)	
	Mean	SD	Mean	SD	Mean	SD
Start to dance (age)	8.32	3.25	8.03	3.13	10.53	3.38
Involvement in dance (years)	9.46	4.61	9.42	4.11	9.80	7.55
Numbers of training per week (count)	3.64	1.52	3.55	1.11	4.33	3.24
Hours of training (hours)	5.81	2.69	5.67	2.07	6.93	5.46
Number of competitions per month (count)	0.58	0.43	0.58	0.43	0.57	0.44
Age (years)	18.12	4.32	17.83	3.82	20.29	6.83
Body fat (%)	21.91	6.39	23.20	5.49	12.12	3.79
BMI (kg/m^2^)	21.56	2.42	21.64	2.46	20.98	2.13
Skeletal Lean Mass (kg)	43.87	6.01	42.41	4.28	54.99	5.67
Fat Free Mass (kg)	46.62	6.36	45.08	4.55	58.32	6.07
Body height (cm)	166.53	7.14	165.01	5.77	178.03	6.14
Body mass (kg)	59.45	8.97	58.51	8.62	66.61	8.62
SEBT_R1 (Result/Leg length)	83.56	9.32	83.59	9.62	83.38	6.99
SEBT_R2 (Result/Leg length)	90.05	9.90	90.02	10.35	90.35	5.68
SEBT_R3 (Result/Leg length)	100.36	14.01	100.71	14.53	97.69	9.14
SEBT_R4 (Result/Leg length)	105.92	15.93	106.34	16.54	102.76	10.05
SEBT_R5 (Result/Leg length)	105.64	16.30	105.93	17.05	103.44	8.90
SEBT_R6 (Result/Leg length)	99.94	15.97	100.46	16.66	96.01	8.63
SEBT_R7 (Result/Leg length)	89.02	14.57	89.36	14.94	86.45	11.46
SEBT_R8 (Result/Leg length)	73.76	11.48	74.79	9.84	66.03	18.63
SEBT_L1 (Result/Leg length)	83.17	9.71	83.19	10.07	83.06	6.66
SEBT_L2 (Result/Leg length)	89.65	9.90	89.46	10.26	91.11	6.68
SEBT_L3 (Result/Leg length)	99.19	14.04	99.36	14.71	97.91	7.52
SEBT_L4 (Result/Leg length)	107.51	16.07	107.94	16.83	104.33	8.23
SEBT_L5 (Result/Leg length)	106.46	17.00	106.57	17.76	105.66	9.92
SEBT_L6 (Result/Leg length)	100.77	17.00	100.66	17.80	101.58	9.29
SEBT_L7 (Result/Leg length)	90.43	14.76	90.46	15.18	90.27	11.58
SEBT_L8 (Result/Leg length)	73.68	11.38	73.72	11.97	73.35	5.50

LEGEND: SEBT—result on Star Excursion Balance Test while standing on the left (L), and right (R) leg in eight directions (1–8).

**Table 2 ijerph-16-03214-t002:** Location of the musculoskeletal (MSK) injury/problem in hip hop dancers over the 3-month study period according to Oslo Sports Trauma Research Center Overuse Injury Questionnaire (OSTRC).

Location	Total		Females		Males	
	F	%	F	%	F	%
Knee	43	42.6%	36	41.9%	7	46.7%
Shoulder	11	10.9%	8	9.3%	3	20.0%
Back	33	32.7%	30	34.9%	3	20.0%
Ankle	14	13.9%	12	14.0%	2	13.3%

**Table 3 ijerph-16-03214-t003:** Correlates of musculoskeletal (MSK) injury and problem in hip hop dancers as obtained by Oslo Sports Trauma Research Center Overuse Injury Questionnaire—crude (non-adjusted) multinomial regression univariate correlations between predictors and criteria (“no MSK problem/injury” as reference value).

Predictors	MSK Problem	MSK Injury
	OR (95% CI)	OR (95% CI)
Dance score	0.6 (0.50–1.47)	1.29 (0.73–2.60)
Involvement in dance (years)	1.04 (0.95–1.14)	1.09 (0.99–1.21)
Numbers of training per week (count)	1.2 (0.91–1.59)	1.07 (0.73–1.56)
Hours of training per week (hours)	1.07 (0.93–1.25)	1.05 (0.87–1.27)
Number of competitions per month (count)	0.94 (0.38–2.31)	1.02 (0.33–3.15)
Age (years)	1.07 (0.99–1.19)	1.05 (0.93–1.19)
Body height (cm)	1.01 (0.96–1.07)	1 (0.94–1.07)
Body mass (kg)	1.01 (0.99–1.06)	1.01 (0.96–1.04)
Body mass index (kg/m^2^)	0.97 (0.83–1.14)	0.98 (0.81–1.21)
Body fat (%)	0.99 (0.93–1.05)	0.99 (0.93–1.08)
Skeletal Lean Mass (kg)	1.01 (0.95–1.08)	0.99 (0.92–1.08)
Fat Free Mass (kg)	1.01 (0.95–1.07)	1 (0.93–1.08)
SEBT_R1 (Result/Leg length)	0.98 (0.94–1.03)	0.96 (0.92–1.02)
SEBT_R2 (Result/Leg length)	0.99 (0.94–1.04)	0.96 (0.91–1.00)
SEBT_R3 (Result/Leg length)	1.03 (0.99–1.07)	0.96 (0.92–1.00)
SEBT_R4 (Result/Leg length)	1.02 (0.99–1.05)	0.95 (0.92–0.99)
SEBT_R5 (Result/Leg length)	1.02 (0.99–1.05)	0.94 (0.92–0.97)
SEBT_R6 (Result/Leg length)	0.95 (0.91–0.97)	0.98 (0.94–1.00)
SEBT_R7 (Result/Leg length)	1.02 (0.98–1.05)	0.98 (0.95–1.01)
SEBT_R8 (Result/Leg length)	0.99 (0.96–1.04)	0.98 (0.95–1.02)
SEBT_L1 (Result/Leg length)	0.99 (0.95–1.04)	0.96 (0.91–1.01)
SEBT_L2 (Result/Leg length)	1.02 (0.97–1.07)	0.96 (0.92–1.01)
SEBT_L3 (Result/Leg length)	1.02 (0.99–1.06)	0.95 (0.91–0.99)
SEBT_L4 (Result/Leg length)	1.01 (0.98–1.04)	0.95 (0.92–0.99)
SEBT_L5 (Result/Leg length)	1.02 (0.99–1.05)	0.96 (0.93–0.99)
SEBT_L6 (Result/Leg length)	1.01 (0.99–1.04)	0.96 (0.93–0.99)
SEBT_L7 (Result/Leg length)	0.98 (0.96–0.99)	0.97 (0.94–1.01)
SEBT_L8 (Result/Leg length)	0.98 (0.94–1.02)	0.97 (0.93–1.01)

LEGEND: Dance score—level of the quality of the dancer, SEBT—result on Star Excursion Balance Test while standing on the left (L) and right (R) leg in eight directions (1–8).

**Table 4 ijerph-16-03214-t004:** Correlates of musculoskeletal (MSK) injury and problem in hip hop dancers as obtained by Oslo Sports Trauma Research Center Overuse Injury Questionnaire—multinomial regression univariate correlations between predictors and criteria (“no MSK problem/injury” as reference value)—adjusted for gender and previous injury occurrence.

Predictors	MSK Problem	MSK Injury
	OR (95% CI)	OR (95% CI)
Dance score	1.1 (0.64–1.89)	1.77 (0.89–3.52)
Involvement in dance (years)	1.04 (0.94–1.16)	1.11 (0.99–1.24)
Numbers of training per week (count)	1.16 (0.87–1.56)	1.02 (0.68–1.52)
Hours of training per week (hours)	1.04 (0.89–1.21)	1.01 (0.83–1.23)
Number of competitions per month (count)	0.79 (0.30–2.09)	0.77 (0.24–2.52)
Age (years)	1.07 (0.96–1.20)	1.04 (0.92–1.19)
Body height (cm)	0.98 (0.92–1.06)	0.97 (0.89–1.06)
Body mass (kg)	1 (0.96–1.05)	1 (0.94–1.06)
Body mass index (kg/m^2^)	0.99 (0.84–1.18)	1.01 (0.82–1.25)
Body fat (%)	1.01 (0.93–1.09)	1.02 (0.93–1.12)
Skeletal Lean Mass (kg)	0.98 (0.89–1.08)	0.96 (0.85–1.08)
Fat Free Mass (kg)	0.98 (0.90–1.07)	0.97 (0.87–1.08)
SEBT_R1 (Result/Leg length)	0.97 (0.92–1.03)	0.96 (0.90–1.01)
SEBT_R2 (Result/Leg length)	0.98 (0.93–1.04)	0.96 (0.90–1.01)
SEBT_R3 (Result/Leg length)	1.04 (0.98–1.08)	0.97 (0.93–1.01)
SEBT_R4 (Result/Leg length)	1.03 (0.98–1.07)	0.96 (0.92–1.00)
SEBT_R5 (Result/Leg length)	1.03 (0.98–1.06)	0.98 (0.94–1.01)
SEBT_R6 (Result/Leg length)	1.04 (0.99–1.08)	0.98 (0.95–1.02)
SEBT_R7 (Result/Leg length)	1.02 (0.99–1.06)	0.98 (0.95–1.02)
SEBT_R8 (Result/Leg length)	1 (0.96–1.05)	0.99 (0.95–1.03)
SEBT_L1 (Result/Leg length)	0.98 (0.94–1.04)	0.96 (0.91–1.01)
SEBT_L2 (Result/Leg length)	1 (0.96–1.06)	0.96 (0.92–1.02)
SEBT_L3 (Result/Leg length)	1.02 (0.99–1.06)	0.96 (0.92–0.99)
SEBT_L4 (Result/Leg length)	1.02 (0.99–1.06)	0.97 (0.93–1.01)
SEBT_L5 (Result/Leg length)	1.03 (0.98–1.06)	0.96 (0.93–0.98)
SEBT_L6 (Result/Leg length)	1.03 (0.99–1.06)	0.97 (0.93–0.99)
SEBT_L7 (Result/Leg length)	1.04 (0.98–1.07)	0.98 (0.94–1.01)
SEBT_L8 (Result/Leg length)	0.98 (0.95–1.02)	0.97 (0.93–1.01)

LEGEND: Dance score—level of the quality of the dancer, SEBT—result on Star Excursion Balance Test while standing on the left (L) and right (R) leg in eight directions (1–8).

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
