# Peer review of "Investigating the Prevalence and Predictors of Injury Occurrence in Competitive Hip Hop Dancers: Prospective Analysis"

_ijerph, 2019, doi:10.3390/ijerph16173214_

Round 1

Reviewer 1 Report

Thank you for the opportunity to review the manuscript titled “Balance capacity is a predictor of injury occurrence in hip hop dance irrespective of previous injury status and gender: prospective analysis”. This is a basic epidemiological study of an under-researched population and thus a useful addition to the literature. The manuscript is relatively well written. However, I do have a few suggestions regarding the analysis and presentation of findings. See specific comments below.

In regard to the sampling strategy, 11 out of 22 dance schools with registered dancers aged >14 years were randomly selected. How many registered dancers aged >14 years are there in Slovenia? How many registered dancers aged >14 years are there in the 11 selected dance schools? These numbers would be useful for the reader to get an idea of the response rate and representativeness. I suggest using the commonplace abbreviation ‘MSK’ for the word ‘musculoskeletal’. Supplementary Table 1 contains important descriptive statistics and should be included as a main table. Rather than reporting mean number of injuries per dancer during the study period, report prevalence and incidence. That is, report the number of injured dancers and the study period (i.e. 3-month) prevalence as the number of injured dancers divided by the number of dancers. Then, report the number of injuries, number of exposure-hours, and exposure-time adjusted injury incidence rates with 95%CI for Poisson rates. Calculate rate ratio with 95%CI for Poisson rates for direct comparison of injury incidence rates (e.g. males versus females). In regard to the regression modelling, please describe the model fitting procedure in more detail. Did the authors use forward selection or backward elimination procedure for the final fitted model? Were the model diagnostics checked? I suspect there will be major issues with multicollinearity as several of the variables would be strongly correlated (e.g. BMI, height and weight). Also, I don’t understand why the authors present three models, typically it will suffice to present crude/unadjusted/univariate estimates and the adjusted estimates from a final fitted model. Lastly, rather than using logistic regression, which results in loss of data, the authors should consider using Poisson regression using the number of injuries as the outcome with the exposure-time as an offset. In addition to the issues surrounding the model fitting procedures mentioned above, there are issues about interpretation. The emphasis on ‘a significant protective effect of proper dynamic balance’ appears to be overstated. In Model 3, only 3 out of 16 SEBT measures were statistically significant (albeit barely so) and the effect sizes are rather miniscule (i.e. 3%-4% lower odds). I doubt that constitutes a minimally clinical/practical important difference. I believe the authors ought to be much more careful and conservative in their interpretation. Having said that, first they need to address the model fitting issues mentioned above. Figure 2 should be converted to a table. Figure 3 is unhelpful and can be omitted.

Author Response

Dear Sir/Madam

Thank you for your review. Please find below how we responded to your comments and where to find amendments in text

Staying at your disposal.

Authors

Thank you for the opportunity to review the manuscript titled “Balance capacity is a predictor of injury occurrence in hip hop dance irrespective of previous injury status and gender: prospective analysis”. This is a basic epidemiological study of an under-researched population and thus a useful addition to the literature. The manuscript is relatively well written. However, I do have a few suggestions regarding the analysis and presentation of findings. See specific comments below.

RESPONSE: Thank you for recognizing the potential of our work. Also, we thank you for the constructive and elaborated comments and suggestions. We followed it and tried to amended the manuscript accordingly. Please see below for detailed responses on your comments.

Staying at your disposal.

Authors

In regard to the sampling strategy, 11 out of 22 dance schools with registered dancers aged >14 years were randomly selected. How many registered dancers aged >14 years are there in Slovenia? How many registered dancers aged >14 years are there in the 11 selected dance schools? These numbers would be useful for the reader to get an idea of the response rate and representativeness.

RESPONSE: Thank you for the suggestion. The details about number of dancers are added. Text reads: "At study baseline, there were 22 dance schools with registered dancers older than 14 years of age in the country, with 345 dancers older than 14 years. The 50% of the schools were randomly selected and invited to participate in the study (e.g. 11 schools with 189 dancers older than 14 years)." (Please see highlighted text in Participants subsection)

 I suggest using the commonplace abbreviation ‘MSK’ for the word ‘musculoskeletal’.

RESPONSE: Following your suggestion the term musculoskeletal is abbreviated when mentioned for the first time, and we used abbreviation MSK later. Thank you.

Supplementary Table 1 contains important descriptive statistics and should be included as a main table.

RESPONSE: Supplementary table 1 is now included as part of the main text (now Table 1). Thank you.

Rather than reporting mean number of injuries per dancer during the study period, report prevalence and incidence. That is, report the number of injured dancers and the study period (i.e. 3-month) prevalence as the number of injured dancers divided by the number of dancers. Then, report the number of injuries, number of exposure-hours, and exposure-time adjusted injury incidence rates with 95%CI for Poisson rates. Calculate rate ratio with 95%CI for Poisson rates for direct comparison of injury incidence rates (e.g. males versus females).

RESPONSE: Thank you for this suggestion. Text is amended accordingly and now reads: “During the course of the study, 64 dancers reported at least one injury, which resulted in a prevalence of 50%. Altogether 101 injuries occurred (95%CI: 82-122), over approximately 10,400 hours of exposure (90 hours of exposure per dancer). In average, each dancer suffered 0.78 injuries (95%CI: 0.61-0.97) across study period of 3 months (0.76 [95%CI: 0.60-0.95] and 0.93 [95%CI: 0.75-1.13], in females and males, respectively), with no significant difference between genders (Mann Whitney Z-value: 0.68, p = 0.52). It equates to rate of 3.12 injuries per dancer per year (e.g. annual incidence of 12%). When expressed with regard to exposure time – hours of dance (HD), the prevalence was 8.7 injuries per 1000HD (95%CI: 6.9-1.1), with somewhat higher prevalence in males than in females (males: 10.33 [95%CI: 5.1-14.2], females: 8.44 [95%CI: 3.5-15.1]). The 17% of dancers reported multiple injuries. Specifically, 10% of dancers reported two, 6% reported three, and 1 dancer (less than 1%) reported four injuries. The 49% of all injuries were time-loss injuries (47% and 55% of time loss injuries in females and males, respectively).  The knee was most frequently injured body location (43% of all reported injuries), then back region (33% of all injuries), and ankle (15%), with similar prevalence across genders (Chi square: 1.17, p = 0.75) (Table 2). “ (please see text after Table 1)

Also, according to your suggestion certain changes are done in Statistics subsection, and text reads: “The normality of the distributions was checked by the Kolmogorov-Smirnov test. For normally distributed variables descriptive statistics included means and standard deviations. Injury rates were reported as total number of injuries per studied period, number of injuries relative to hours of exposure - dance hours (with 95%CI for Poisson rates) Irrespective of OSTRC specific graduation which differs MS-problem from MS-injury, in the following text the MS-problem (OCTRC scores from 1-40) and MS-injury (scores of >40) were collectively considered as “injury” if not specified otherwise.” (please see subsection on Statistics)

In regard to the regression modelling, please describe the model fitting procedure in more detail. Did the authors use forward selection or backward elimination procedure for the final fitted model? Were the model diagnostics checked? I suspect there will be major issues with multicollinearity as several of the variables would be strongly correlated (e.g. BMI, height and weight). Also, I don’t understand why the authors present three models, typically it will suffice to present crude/unadjusted/univariate estimates and the adjusted estimates from a final fitted model. Lastly, rather than using logistic regression, which results in loss of data, the authors should consider using Poisson regression using the number of injuries as the outcome with the exposure-time as an offset. In addition to the issues surrounding the model fitting procedures mentioned above, there are issues about interpretation.

RESPONSE: We must agree that the regression analyses were not sufficiently explained, and actually all concerns raised in previous comment are therefore understandable. Specifically, in this study we didn’t calculate the multivariate regressions but calculated univariates associations between each predictor. Namely, we were aware about the possible collinearity among predictors you have specified in your comment as well, and therefore avoided multivariate calculations, especially since multinomial regression we have used doesn’t include “forward” and “backward” models you have also specified. Amended text now reads: “The association between the studied predictors and outcome (OSCTRC defined presence of MSK-problem and MSK-injury) was evaluated by univariate multinomial regression calculation for a multinomial criterion based on OSTRC scale (0 – absence of MSK problem/injury, 1 – MSK-problem, 2 – MSK-injury), with the absence of a problem/injury as the referent value in the multinomial regression calculation. The odds ratio (OR) with the corresponding 95% confidence interval (95% CI) was reported. Multinomial regression calculations included nonadjusted regression correlations, and correlations adjusted for gender and injury reported at baseline.“ (Please see subsection Statistics)

Next, although we considered Poison calculations the problem is in the fact that we observed the “multinomial” criterion and “bivariate” injury occurrence which would be useful in Poison’s calculations. Therefore, we retained the multinomial regressions with three “criterion responses” (non injury/problem – MSK problem – MSK injury; with the first one as reference value).

According to your suggestion in this version of the manuscript we included only two sets of regression calculations; the non-adjusted (please see Table 3 in the revised version of the manuscript), and correlations adjusted for gender + previous injury status (Table 4). Also, titles of the Tables are amended accordingly. Thank you.

The emphasis on ‘a significant protective effect of proper dynamic balance’ appears to be overstated. In Model 3, only 3 out of 16 SEBT measures were statistically significant (albeit barely so) and the effect sizes are rather miniscule (i.e. 3%-4% lower odds). I doubt that constitutes a minimally clinical/practical important difference. I believe the authors ought to be much more careful and conservative in their interpretation. Having said that, first they need to address the model fitting issues mentioned above.

RESPONSE: We must agree that our protective effects of dynamic balance may look as overstated, and therefore we tried to modify the parts of the text where protective effects of balance were highlighted. In the following text we specified several instances where manuscript is amended accordingly (please see underlined parts of the text.

“One of the most important findings of this study is related to the relationship between balance capacity and injury risk in hip hop dancers. In short, dancers with better balance were less likely to suffer injury during the observed period of time. Although the clinical importance of these findings is probably relatively small, this is one of the first studies which prospectively observed balance as predictors of injury in dance, and from our perspective these results deserve attention.” (please see beginning of the subsection 4.2) “Apart from the logical and expected correlation between previous injury status and injury occurrence (e.g., a higher risk for injury in those dancers who suffered an injury previously), this study demonstrated reduced risk for injury in dancers with better dynamic balance” (please see Conclusion) “While the SEBT requires no special equipment and is generally cheap and not time consuming, we may highlight its applicability in the evaluation of balance in various dance activities and the potential applicability in identification of dancers at specific risk for injury occurrence” (please see Conclusion)

Most importantly, the title of the paper is changed and now reads “Investigating the prevalence and predictors of injury occurrence in competitive hip hop dancers: prospective analysis” (previously: “Balance capacity is a predictor of injury occurrence in hip hop dance irrespective of previous injury status and gender: prospective analysis)

Figure 2 should be converted to a table. Figure 3 is unhelpful and can be omitted. 

RESPONSE: Amended accordingly. Figure 2 is converted in Table (please see Table 2), while Figure 3 is omitted as suggested.

Reviewer 2 Report

The manuscript is an interesting and novel object of study. It is of sufficient quality to be published in the journal, although certain aspects of the document should be improved.

- Improve keywords: they should not be included in the title of the manuscript
- Participants: Why haven't the small number of male athletes been eliminated from the participating sample? The objective is not to compare according to gender. It is very likely that the results were not statistically significant due to the small number of male participants. The authors mention this in the limitations of the study.
- The first paragraph of the discussion is much more similar to conclusions of the study that confirm or not the hypothesis raised. I would begin by recalling the objective of the research.
- Add possible future lines of research that complement and improve the found results and practical applications of the results.

Author Response

The manuscript is an interesting and novel object of study. It is of sufficient quality to be published in the journal, although certain aspects of the document should be improved.

RESPONSE: Thank you for recognizing the quality of the manuscript. Also, we are particularly grateful for your constructive suggestions. We tried to follow it and amended the manuscript accordingly. In the following responses, we indicated the specific changes we made. Staying at your disposal.

- Improve keywords: they should not be included in the title of the manuscript

RESPONSE: Keywords are changed. It now include terms not used in the manuscript (e.g. sport, prevalence, risk factors, protective factors, OSTRC) Thank you.

- Participants: Why haven't the small number of male athletes been eliminated from the participating sample? The objective is not to compare according to gender. It is very likely that the results were not statistically significant due to the small number of male participants. The authors mention this in the limitations of the study.

RESPONSE: Indeed, the studied sample included only the small number of males, but we did not compare the genders as you stated. However, we were of the opinion that „controlling for gender“ in regression analyses was an appropriate approach, while results allowed us to highlight some important aspects of the injury occurrence in males and females (although we did not compare it statistically, as you said). All these aspects are now more precisely explained in the Limitation subsection. The text reads: “The first study limitation is related to the unequal number of female and male dancers. Indeed, the relatively small number of males in the study limits the generalizability of the results. However, this is the situation in Slovenia where the sample was drawn from, and we did not aim to study the differences between genders. Also, despite the small number of studied male dancers we believe that study highlighted some important characteristics of injury occurrence in males (i.e. shoulder as specific location of injury). Therefore, in future studies, special attention should be placed on male hip hop dancers.” (Please see 1st paragraph of the Limitations subsection). Thank you!

- The first paragraph of the discussion is much more similar to conclusions of the study that confirm or not the hypothesis raised. I would begin by recalling the objective of the research.

RESPONSE: Amended accordingly. In this version of the manuscript, the discussion starts with study objectives and the text reads: “This study aimed to prospectively evaluated injury occurrence, and specific factors associated with injury occurrence in hip hop dancers. In general, the prevalence of injury is high, and we evidenced some specific correlates of injury occurrence. The most important findings will be discussed in the following text.” (Please see 1st paragraph of the Introduction). Meanwhile, the hypothesis is moved to the conclusion section (Please see the end of 1st paragraph in the Conclusion). Thank you.

- Add possible future lines of research that complement and improve the found results and practical applications of the results.

RESPONSE: We are particularly grateful for this suggestion. The lines for future investigations is now included in the Conclusion section. Text reads “This study evaluated some specific predictors of injury in hip hop. In future studies other potentially important factors of influence on injury occurrence should be evaluated. In doing so special attention should be placed on those abilities and characteristics which could be potentially related to injury occurrence of some specific body regions (i.e. shoulder in male dancers, lower back in females). Also, knowing the differences in dance styles and physiological- and MSK-demands in various hip hop dance disciplines, the style-specific analyses in this sport is warranted.” (Please see last paragraph of the Conclusion section. Thank you!)